# Lipid Players of Cellular Senescence

**DOI:** 10.3390/metabo10090339

**Published:** 2020-08-21

**Authors:** Alec Millner, G. Ekin Atilla-Gokcumen

**Affiliations:** Department of Chemistry, University at Buffalo, The State University of New York (SUNY), Buffalo, NY 14260, USA; alecmill@buffalo.edu

**Keywords:** lipids, senescence, sphingolipids, fatty acids, glycerolipids

## Abstract

Lipids are emerging as key players of senescence. Here, we review the exciting new findings on the diverse roles of lipids in cellular senescence, most of which are enabled by the advancements in omics approaches. Senescence is a cellular process in which the cell undergoes growth arrest while retaining metabolic activity. At the organismal level, senescence contributes to organismal aging and has been linked to numerous diseases. Current research has documented that senescent cells exhibit global alterations in lipid composition, leading to extensive morphological changes through membrane remodeling. Moreover, senescent cells adopt a secretory phenotype, releasing various components to their environment that can affect the surrounding tissue and induce an inflammatory response. All of these changes are membrane and, thus, lipid related. Our work, and that of others, has revealed that fatty acids, sphingolipids, and glycerolipids are involved in the initiation and maintenance of senescence and its associated inflammatory components. These studies opened up an exciting frontier to investigate the deeper mechanistic understanding of the regulation and function of these lipids in senescence. In this review, we will provide a comprehensive snapshot of the current state of the field and share our enthusiasm for the prospect of potential lipid-related protein targets for small-molecule therapy in pathologies involving senescence and its related inflammatory phenotypes.

## 1. Introduction

### Lipid Diversity and Function

Organisms contain thousands of distinct lipid molecules with diverse functions in fundamental cellular processes. This diversity is achieved through the incorporation of one or multiple aliphatic chain(s), which can vary in length and unsaturation, to different polar backbones or headgroups [1] via the coordinated action of a network of enzymes that synthesize, metabolize, and transport lipids [2]. The lipid content of mammalian cells is classified into different groups, including fatty acids, glycerolipids, phospholipids, sphingolipids, and sterols [1]. Lipids have three general functions, all of which are crucial for cell survival and homeostasis [1].

First, polar lipids, with their amphipathic nature, form the cellular membranes. The hydrophobic moieties self-associate while the hydrophilic head groups interact with the aqueous environment. This phenomenon also allows compartmentalization of organelles within the cell, enabling segregation of chemical reactions and improving efficiency of biochemical pathways (Figure 1A) [3]. The plasma membrane is primarily composed of phospholipids, cholesterol, and glycolipids. The length and unsaturation of the fatty acyl tails directly influences the fluidity of the membrane by affecting how closely the molecules can pack against each other and other biomolecules [4]. The composition of the lipid bilayer also has a profound effect on the interaction and function of membrane proteins [5]. Local lipid composition of the plasma membrane is also critical for the recruitment of certain proteins to these sites, mainly utilizing hydrophobic and electrostatic interactions [6].

Second, neutral lipids such as cholesterol esters and triacylglycerols are mainly used for storing excess fatty acids. These molecules reside within lipid droplets, endoplasmic reticulum-derived organelles, and provide building blocks for membranes or substrates for energy metabolism (Figure 1B) [7]. The biogenesis of lipid droplets is activated at high intracellular lipid levels, resulting in the production of these cytosolic organelles that are composed of a neutral lipid core and enclosed with a phospholipid monolayer [7].

Lastly, lipids and their metabolites can trigger numerous physiological responses and are involved in many fundamental cellular processes. They can bind to various proteins, activate signaling cascades, and elicit a specific cellular response. One such example is the binding of diacylglycerol (DAG) to protein kinase C, which can affect a broad range of cellular programs and adhesion, migration, and proliferation (Figure 1C) [8,9]. Hydrolysis of phosphatidylinositol 4,5-bisphosphate releases inositol-1,4,5-trisphosphate and DAG (Figure 2A). DAG acts as an allosteric activator of protein kinase C, and inositol-1,4,5-trisphosphate binds to calcium channels, modulating intracellular calcium [10]. Spikes in calcium mediate the localization of protein kinase C to membranes with DAG [11]. Upon recruitment to the membrane, protein kinase C is activated and undergoes conformational changes, exposing DAG binding sites [12,13]. This signaling cascade is critical for cellular homeostasis; hence, dysregulation of DAG-dependent protein kinase C activity has been implicated in various diseases including cancer [14] and cardiac disease [15].

Sphingolipids also act as highly bioactive signaling molecules [16]. Accumulation of ceramides and sphingosine can lead to an anti-proliferative or pro-apoptotic response [17]. However, sphingosine can instead be converted to sphingosine-1-phosphate, which promotes cell growth and proliferation (Figure 2B) [18,19]. The interplay between different sphingolipids and how cells maintain a balanced sphingolipid pool is not fully understood, but it is well-established that disrupted sphingolipid levels are observed in numerous diseases [16,20]. Similar to the pro-death and survival roles of different sphingolipids, the products of arachidonic acid in the eicosanoid pathway can exhibit opposing bioactivities. Prostaglandin E2 (PGE2) has an immunosuppressive effect on infected macrophages, whereas prostaglandin D2 (PGD2) stimulates the immune response (Figure 2C) [21].

Whether through their structural or signaling roles, regulation of lipids has been linked to a plethora of cellular programs and diseases. Recent reviews highlight that regulation of specific lipids is required for fundamental processes such as cellular division [22] and cell death [16,23]. These key roles require tight control of lipid biosynthesis and metabolism in order to maintain cellular homeostasis [16,24]. As such, dysregulation of enzymes involved in lipid metabolism has been linked to various diseases, including cancer [25], neurogenerative diseases [16], and aging [26]. Lipid processing enzymes are ideal targets for small-molecule perturbations with potential application in developing novel, lipid-based therapeutic approaches. However, such approaches require knowledge of specific lipid species and pathways that are altered in a particular phenotype. Hence, furthering our knowledge of lipid function in cellular processes is not only critical for understanding the chemistry and biology of these molecules but also for the identification of new therapeutic targets. With advancements in the omics fields, especially metabolomics and transcriptomics, these lipid-related therapeutic targets and novel structure-specific functions for lipids can be identified for various diseases and cellular processes.

One cellular process that is fundamentally linked to extensive lipid and membrane remodeling is senescence [27]. Cellular senescence is a process by which a cell enters a state of permanent growth arrest [27]. These cells remain metabolically active and can be characterized by morphological changes, altered gene expression and protein synthesis, as well as an activated secretory phenotype [27]. Senescence has been shown to be a major driver in aging [28] and cancer recurrence [29]; thus, developing a better understanding of the process has important implications in various diseases. Lipidomics [30,31,32] and transcriptomics [30,33,34] shed light onto the lipid components and pathways that are involved in this process, how cells might achieve this state, and how they might interact and impact their environment through the activated secretory phenotype. In this review, we will discuss lipid classes that have been associated with senescence with an emphasis on their potential roles during this process.

## 2. Mechanisms and Functions of Cellular Senescence

The senescence program and the associated cell cycle arrest can be induced through various stimuli. Nearly 60 years ago, Hayflick and Moorhead showed that primary cells enter a non-dividing, senescent state upon continuous culturing, reaching their proliferative capacity [35]. This phenomenon is now termed replicative senescence. Later, it was found that other stimuli, such as active oncogenes [36] and chemotherapy [37], can also induce senescence prematurely (Figure 3). These processes are broadly defined as stress-induced senescence [38]. In addition to cell cycle arrest, senescent cells become enlarged with increased lysosomal activity, and they have a unique secretome known as the senescence-associated secretory phenotype (SASP, Figure 3) [39]. Many of these markers individually are not unique to senescence; thus, a combination of these markers is often used to characterize the senescent phenotype.

Stable cell cycle arrest is key for senescence and is achieved by the inhibition of cyclin/cyclin-dependent kinases by the p53/p21 and p16/retinoblastoma protein (RB) pathways (Figure 3). Initial cell cycle arrest is mediated by p21, which prevents cells from entering S-phase under stress conditions, i.e., DNA damage [40]. If the cell cannot reenter the cell cycle, upregulated p16 activates RB, which is thought to maintain the stable cell cycle arrest associated with senescence (Figure 3) [41].

Irregular and enlarged cellular morphology are hallmarks of senescent cells in culture. Cell enlargement is a result of the activation of the mTOR pathway during senescence [42]. Increased mTOR signaling promotes multiple processes including nutrient uptake and protein and lipid biosynthesis, which contributes to increased cell size [43]. The changes in cell shape are largely attributed to a combination of cytoskeleton rearrangements and signaling from the Unfolded Protein Response [27,44]. These morphological changes can easily be observed and measured in cell culture but are challenging to detect in vivo.

Another characteristic of the senescent state is the increased lysosomal content and upregulated lysosomal proteins. Among these proteins is the most commonly used senescence marker, senescence-associated β-galactosidase (SA-β-gal). The increased abundance of this enzyme is due to high expression of galactosidase beta 1 [45] most likely due to the increased lysosomal content of senescent cells. However, whether increased SA-β-gal activity has a specific function in senescence is unknown.

Alongside the widespread changes in intracellular protein and gene expression, senescent cells secrete cytokines, growth factors, proteases, chemokines, other extracellular matrix components, and small molecules such as prostaglandin E2 into their environment [46]. This secretory phenotype, SASP, is highly heterogeneous across different cell types [47], senescent-inducing stimuli [48], and during different stages of senescence [49]. The secreted molecules affect surrounding cells by their interactions with the cell surface receptors, leading to various responses [50,51]. Due to the heterogeneity of the secretome and available receptors, the SASP has been linked to many cellular processes and diseases.

Although first associated with organismal aging, senescence is linked to several cellular processes and diseases [49]. It has been suggested that senescence has evolved as a mechanism to overcome cancer formation via growth arrest and pro-inflammatory factors in the SASP [52]. The released SASP components exert their tumor-suppressive effects by recruiting different types of immune cells to the tumor environment, resulting in the elimination of senescent and damaged cells [53,54]. For instance, senescent cells have been shown to restrain tumorigenesis in a mouse model of hepatocellular carcinoma [55]. Similarly, the immune response caused by activated SASP plays a role in wound healing. Among the released components is the AA isoform of platelet-derived growth factors, which promote differentiation and, ultimately, wound closure [56]. Accumulation of senescent cells has been observed at sites of wounding and are implicated to aid in tissue remodeling [49]. Senescent cells appear to be important for overall tissue healing, as clearing senescent cells from the wound site delays healing [57]. However, it is important to note that studies have also shown tumorigenic activity of senescent cells, where invasion and metastasis were promoted [39,58], which suggests the tissue- and context-dependent effects of senescent cells and the SASP.

As humans age, the risk of many diseases, including diabetes, arthritis, and cardiovascular diseases, increases. There is accumulating evidence linking sex-specific differences in lipid and sterol-derived hormone metabolism to aging and related diseases [59,60]. Senescent cells accumulate with age and are thought to contribute to these diseases, largely through the SASP and disruption of tissue structure and functioning. Accumulation of senescent cells could compromise the ability of tissue to regenerate or function properly either through growth arrest or altering the environment via the SASP [61]. Specifically, the SASP components interleukin 6 and 8 may allow senescent fibroblasts to act as pro-inflammatory cells, which stimulates tissue fibrosis [61,62]. Prostaglandin E2, another common SASP component, is involved in the immune response [63]. It is highly upregulated during senescence [64] and is released to the extracellular environment [65,66]. Furthermore, exposure of cells to prostaglandin E2 accelerates the development of senescent markers distinct from those caused by oxidative stress [67].

Senescent cells pose both beneficial and detrimental effects in vivo, and we still do not yet know exactly why. This highlights the gap in our current knowledge on fully defining the molecular components of this process and understanding the mechanism by which senescent cells affect their environment. As exemplified above, senescence and the SASP require extensive lipid and membrane remodeling. A better understanding of the role lipids play in remodeling could fill this gap in knowledge and uncover potential lipid-related proteins that can be targeted to influence senescence.

## 3. Various Lipid Classes Have Been Linked to Senescence

Given the vast membrane remodeling in senescence, lipids play a key role in establishing and maintaining this process. Considerable research efforts have been made towards understanding the proteins involved in senescence including the activation of p16 and/or p53 and the distinct secretory phenotype, SASP [46]. However, there is a major gap in understanding the identity, functional role, and regulation of lipids in this process. Integrating transcriptomics and lipidomics, we recently showed that both expressions of lipid-regulating genes and lipid levels are highly regulated, and a greater proportion of lipid-related genes are differentially expressed in senescent cells as compared to the rest of the transcriptome, suggesting that, at a global level, the regulation of lipids is central to this process [30]. In this section, we review the lipid classes that have been associated with senescence with an emphasis on their mechanism of action when available.

### 3.1. Fatty Acids

Fatty acids are among the simpler lipids that serve as sources of energy and building blocks of complex lipids. Cells obtain fatty acids either by de novo biosynthesis or by the uptake of exogenous fatty acids. At the organismal level, the accumulation of fatty acids and the changes in their unsaturation have been associated with aging (reviewed in [68]), specifically the levels of plasma poly- and mono-unsaturated fatty acids increase with age in healthy individuals [69].

Studies have suggested senescent cells and aged tissues primarily use de novo synthesis to maintain fatty acid levels by increasing fatty acid synthase (FASN) levels [70]. During oncogene-induced senescence, however, FASN is inhibited suggesting that these cells rely more on fatty acid uptake [71]. As a result, it is plausible to envision fatty acid transporters and enzymes that process fatty acids can also be functionally involved in senescence, in addition to these lipids themselves.

Carnitine palmitoyl transferase 1 (CPT1), a transmembrane enzyme located on the outer membrane of the mitochondria, has been linked to cellular senescence. CPT1 converts long-chain fatty acyl-CoA to long-chain acylcarnitine and mediates their transport to the inner membrane where acyl carnitines can be broken down via β-oxidation [72,73]. As a result, the activity of this enzyme is not only important for lipid metabolism but also for maintaining mitochondrial membrane dynamics, reactive oxygen species, and consequently DNA damage and lipid peroxidation [73,74].

The levels of CPT1 are regulated by peroxisome proliferator-activated receptor (PPARα), acetyl-CoA carboxylase 1 (ACC1) activity, and malonyl CoA levels. Chen et al. showed that depletion of PPARα resulted in low CPT1C expression and induced cellular senescence in cancer cell lines, while its overexpression promoted proliferation [75]. ACC1 catalyzes the first step in de novo fatty acid biosynthesis, ATP-dependent carboxylation of acetyl-CoA to malonyl-CoA. ACC1 can stimulate CPT1 activity and increase the influx of long-chain fatty acids into the mitochondria [76,77]. Seok et al. found that CPT1 expression and fatty acid oxidation were increased in senescent human placenta-derived mesenchymal stem cells as compared to their proliferating counterparts, likely through the inactivation of ACC1 [78]. They also showed that mitochondrial dysfunction, the levels of reactive oxygen species, and senescence were reduced when CPT1 was inhibited [78], suggesting that CPT1 activity and fatty acid breakdown might play roles in maintaining mitochondrial function in senescence. Overall, these studies suggest the involvement of fatty acid and their metabolism in cellular senescence occur through a complex mechanism that seems to be dependent on tissue type and developmental stage.

### 3.2. CD36

CD36 (cluster of differentiation 36), a membrane receptor found to assist in fatty acid uptake and overall lipid turnover, has been implicated in fatty acid metabolism [79,80]. CD36 can undergo various post-translational modifications including ubiquitination [81], glycosylation [82], and palmitoylation [83]. These modifications are likely involved in trafficking CD36 between membranes to promote organelle-specific localization and affect its function [84]. CD36 has a high affinity for long-chain fatty acids (fatty acids with 18 carbons and longer) [85,86]. The uptake of these fatty acids can have pleiotropic effects on lipid biosynthesis and beta oxidation [87], and intracellular signaling [88]. In addition to fatty acids, CD36 recognizes phospholipids [89], diacylglycerol [90], and high- and low-density lipoproteins [91]. This receptor has been found on various cell types [82] and has been shown to mediate cell-specific responses such as angiogenesis inhibition in endothelial cells [92], inflammation in macrophages [93], and platelet aggregation and activation [94]. Variation in the expression of CD36 has been associated with several diseases and related states in humans including Alzheimer’s disease [95] and inflammation [96].

Transcriptomic studies linked the changes in CD36 expression to senescence [30,33]. Our study on replicative senescence showed that the increase in CD36 expression in late-passage human fibroblast cell lines constituted one of the most profound changes in lipid-related transcripts as compared to their early passage counterparts. Intriguingly, overexpression of CD36 alone induced a senescence-like phenotype in proliferating young cells, and culturing proliferating fibroblasts in the growth media of CD36-overexpressing cells induced premature senescence [30]. Based on these findings, we envision that CD36 upregulation might contribute to membrane remodeling and activated SASP during replicative senescence. Concordant with these findings, Chong et al. showed increased CD36 expression during various types of senescence programming (i.e., replicative, oncogene-, and therapy-induced senescence) [33]. Upon CD36 expression, they found that SASP components interleukin 6 and 8 were increased [33]. At this time, it is yet unclear if the involvement of CD36 is through membrane remodeling and/or intracellular signaling mediated through fatty acid uptake. Further studies are needed to shed light onto this exciting research venue.

### 3.3. Glycerolipids

DAGs and triacylglycerols (TAGs) are the major glycerolipids in the cellular lipidome. DAGs can function as signaling lipids [13] and are precursors for glycerophospholipids [97]. TAGs, on the other hand, are one of the few lipid classes that are mainly non-membrane associated and are stored within lipid droplets [7]. The primary function of these organelles is the storage of excess lipids in cells, and recent studies have shown that they play active roles in lipid accumulation and metabolism (reviewed in [7]).

Untargeted lipidomics in fibroblasts undergoing replicative senescence showed that the accumulation of TAGs and lipid droplets constituted the majority of lipid-related changes during this process [32]. Specifically, TAG species that accumulated during senescence consisted of polyunsaturated fatty acids and were stored within lipid droplets. Another study showed that lipid droplets and enzymes involved in breaking down reactive lipids were upregulated during therapy-induced senescence, suggesting that lipid peroxidation may drive increased lipid uptake and metabolism [34]. Studies have reported accumulation of lipid droplets under cellular stress conditions both in vivo [98] and in vitro [99,100] and suggested protective roles for these structures. It is possible that the accumulation of lipid droplets also plays a protective role during senescence. By diverting unsaturated fatty acids to TAGs and storing them in lipid droplets, cells are able to sequester these highly reactive species that are prone to oxidation and forming lipid peroxides away from membranes and limit membrane damage under oxidative stress during senescence.

### 3.4. Phospholipids

In addition to their membrane-forming properties [4], phospholipids and their metabolic products regulate various processes and signals. Phosphatidylserine is normally located on the inner leaflet of the membrane but can be exposed on the cell surface by scramblases [101]. Exposure of phosphatidylserine on the surface serves as a signal for macrophages to engulf the cell [102]. Phospholipids can also be hydrolyzed by phospholipases to yield free fatty acids and lysophospholipids [103]. The free fatty acid is released and available for metabolism, and the remaining lysophospholipid can play an important role in cellular processes [104]. For instance, lysophosphatidic acid binds to receptors and influences various processes including proliferation [105] and cell survival [106]. Lysophosphatidic acid has been shown to participate in cancer progression and invasion [107] as well as cardiovascular disease, in which the effect can be protective [108] or harmful [109] depending on the expression of lysophosphatidic acid-binding receptors. Lysophosphatidylcholine is also involved in multiple signaling pathways involved in oxidative stress [110] and inflammation [111], and high lysophosphatidylcholine levels disrupt mitochondrial integrity, inducing apoptosis [112].

Although phospholipid signaling is involved in processes related to senescence, oxidative stress, and inflammation, changes to overall phospholipid levels have yet to be functionally linked to establishing or maintaining senescence in mammalian cells. Proteomic analysis revealed upregulation of several phospholipases in stress-induced senescence [34]. Flor et al. observed increased lipid peroxidation, lipid aldehydes, and aldehyde quenching/metabolizing enzymes in stress-induced senescence [34], implicating that the aldehyde end-products of lipid peroxidation are important mediators of this process. Furthermore, addition of the reactive aldehyde 4-hydroxy-2-nonenal can induce senescence, while reducing aldehydes can hinder the effects of senescence inducers [113]. Consistent with the role of lipid droplets accumulation during senescence, as we discussed above, these studies further implicate the involvement of lipid-induced membrane damage in senescence.

### 3.5. Sphingolipids

Sphingolipids are highly bioactive lipids. They play structural roles in cellular membranes and function as signaling molecules [114]. Although sphingolipids share a sphingoid backbone and are structurally related (Figure 4), the biological properties of these sphingolipids differ greatly. Ceramides are central lipids in the sphingolipid biosynthesis, and their levels are regulated by de novo biosynthesis, breakdown of complex sphingolipids, and the salvage pathway [16]. The incorporation of serine into palmitoyl CoA initiates de novo biosynthesis resulting in dihydroceramides and ceramides via the activity of ceramide synthases (Figure 4). Ceramides can then be hydrolyzed to form sphingosine, which can be phosphorylated to produce sphingosine-1-phosphate (S1P) [16]. Sphingosine and S1P have shown to be involved in numerous processes regulating cell fates, with sphingosine having pro-death properties [115] and S1P promoting proliferation [116] (Figure 4).

Ceramides can form complex sphingolipids, sphingomyelins, and glycosphingolipids. Both sphingomyelins and glycosphingolipids serve as structural lipids and are mainly found on the plasma membrane [117,118]. Studies have suggested that sphingomyelin colocalizes with cholesterol [119] and is important for membrane cholesterol homeostasis [120]. Studies have suggested that sphingomyelin is also important for some protein–membrane association [121,122]. Glycosphingolipids have been shown to play important roles in intracellular trafficking [123] and recognition [124] and have been found to play a role in inflammation [125,126]. Dysregulation of sphingomyelin or glycosphingolipid pathways can contribute to changes in ceramide levels, which are extensively involved in cellular proliferation and are cytotoxic at high levels [16].

Sphingolipids are one of the most heavily studied lipids in senescence. Various studies have correlated increased ceramide levels with replicative [127,128] and stress-induced [31] senescence. Furthermore, multiple studies have shown that young proliferating cells grown in the presence of low concentrations of C6-ceramide (<20 μM) can undergo growth arrest [129] and show increased expression of SA-β-gal [130] analogous to senescence. However, treatments with higher concentrations of ceramide induce cell death, likely through disruption of mitochondrial dynamics and release of cytochrome C [16]. These studies suggest that ceramides play a functional role in senescence; however, their role in this process is still unclear and likely to differ from their role in apoptosis.

Sphingosine kinase 1 and 2 (SK1 and SK2) catalyze the phosphorylation of sphingosine to generate S1P (Figure 4). S1P is an important bioactive metabolite that has been shown to promote proliferation [116] and downregulation of SK1, and decreased S1P levels have been associated with reducing proliferative capacity, hence accelerating senescence [131,132]. This accelerated senescence was reversed with cotreatment with S1P and fumonisin B1, an inhibitor of ceramide synthase, suggesting that cellular senescence accelerated by low SK1 levels might be linked to increased ceramide levels as well as decreased S1P levels [131].

S1P binds to multiple proteins including human telomerase reverse transcriptase (hTERT), several G-protein coupled receptors, and histone deacetylase (reviewed in [133]). Binding of hTERT to S1P is suggested to mimic hTERT phosphorylation, enhancing its stability and thus helping to regulate the telomere damage associated with senescence [134]. Binding of S1P to sphingosine-1-phosphate receptor subtype 2 (S1PR2) showed an increase in the levels of senescent endothelial cells, and its activation is associated with increased pro-inflammatory cytokines and lipid mediators [135]. Although the exact role of S1P in senescence is not yet elucidated, it is possible that both S1P levels and the expression levels of receptors that it binds to may play an important role in growth arrest and the associated secretory phenotype.

Ceramide can be generated through the salvage pathway by the degradation of sphingomyelin by sphingomyelinases (Figure 4). Two sphingomyelinase isoforms exist, acid sphingomyelinase and neutral sphingomyelinase (N-SMase), with N-SMase activity having been associated with senescence [133]. N-SMase is localized primarily at the plasma membrane [136]. Changes in N-SMase levels have been reported in aging tissues [137]. A recent study has suggested that upregulation of N-SMase and ceramide biosynthesis contributed to the production of extracellular vesicles and prevented excessive inflammatory response [138]. However, the exact mechanism of how N-SMase activity contributes to senescence, whether it affects membrane dynamics or generates ceramides at certain membrane locales, is not known and is likely to be tissue specific. Based on these studies investigating the role of ceramides and S1P, it is plausible to envision that it is the interplay and balance between pro-death and survival sphingolipids that is important for the initiation and maintenance of the senescence phenotype.

Deoxyceramides are atypical sphingolipids that are synthesized via the incorporation of alanine instead of serine to palmitoyl-CoA by serine palmitoyl transferase, producing deoxysphinganine (Figure 4). Much like the canonical sphinganine, deoxysphinganine can be acylated with fatty acids of various lengths, resulting in deoxyceramides. Due to the lack of the 1-hydroxyl group, however, deoxyceramides cannot be incorporated into more complex sphingolipids or be degraded by the canonical salvage pathway [139]. As compared to ceramides, deoxyceramides are more hydrophobic and less likely to self-organize into bilayers. As such, the changes in their levels could impact membrane packing and integrity [140]. At the organismal level, deoxyceramides accumulate in tissues of aged mice as compared to a young control group [141]. We have recently shown that deoxyceramides were depleted in stress-induced senescence, and importantly, increasing the levels of deoxyceramides reduced the number of senescent cells, suggesting a functional involvement [31]. These species might act as molecular intermediates through multiple different mechanisms during therapy-induced senescence. The changes in the levels of these species can affect the levels of other sphingolipids since they cannot form sphingomyelins and glycosphingolipids. It is also possible that they hinder sphingolipid signaling or affect membrane dynamics due to their increased hydrophobicity.

## 4. Concluding Remarks

Lipids are key small-molecule metabolites with crucial cellular functions. There is a growing appreciation of the various roles they play in numerous cellular processes as structural and signaling molecules, and molecules that mediate changes in membrane properties. Senescence is a cellular state at which proliferation is halted, yet the cells are still metabolically active. Mainly studied in tissue culture, cellular senescence has been linked to aging, cancer formation and progression, development, and other diseases at the organismal level. Downregulation of cell cycle regulating proteins is a hallmark of senescence, and lipid-related pathways are significantly enriched among the upregulated transcripts [32]. This global upregulation of lipid biosynthesis and metabolism could be related to increasing membrane demand due to enlarged size and the activated secretory phenotype involved in the release of inflammatory protein and small-molecule components to the cellular environment. Homing in on this global activation of lipid metabolism, certain lipid families have been associated with different types of senescence programming with roles in cell cycle arrest, inflammatory phenotypes, activation of SASP, and response to oxidative stress. We are just scratching the surface of understanding the regulation and function of these lipid species during cellular senescence, and much exciting work remains to be done along these lines. An improved mechanistic understanding of the regulation and function of these molecules will put forward lipid-related druggable protein targets and pave the way to new therapeutic strategies to perturb senescence and its associated inflammatory effects in numerous diseases.

## Figures and Tables

**Figure 1 metabolites-10-00339-f001:**
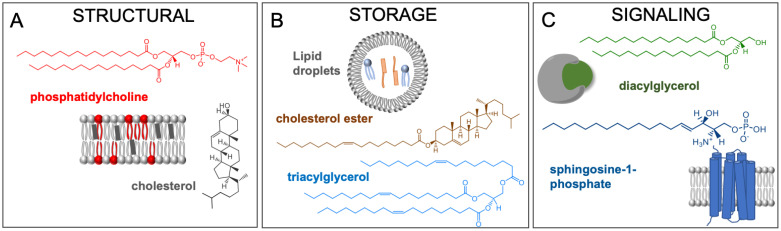
Different roles of lipids. (**A**) Phosphatidylcholine and cholesterol are major structural lipids. (**B**) Cholesterol esters and triacylglycerols are used to stored excess fatty acids in cells. They are stored within lipid droplets and endoplasmic reticulum-derived vesicular structures. (**C**) Diacylglycerol and sphingosine-1-phosphate are signaling lipids. Diacylglycerols affect protein kinase C activity, and sphingosine-1-phosphate promotes proliferation through its interactions with cell surface receptors.

**Figure 2 metabolites-10-00339-f002:**
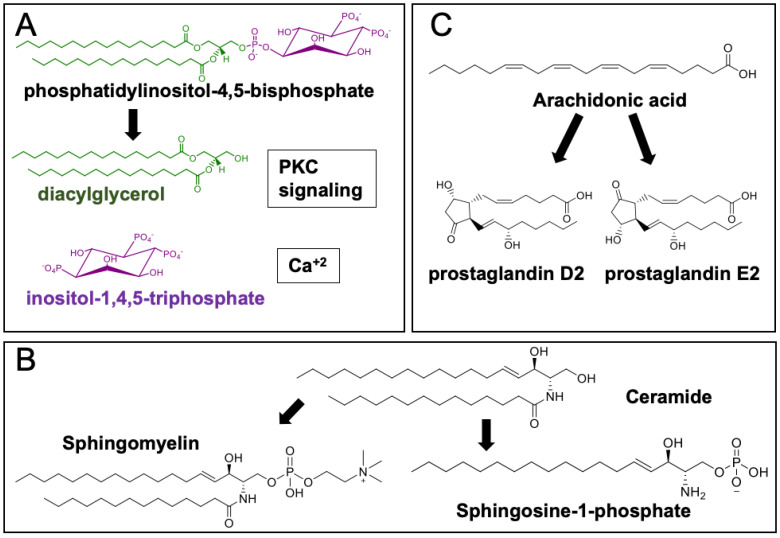
Metabolites of a particular lipid can have different bioactivities. (**A**) Hydrolysis products of phosphatidylinositol 4,5-bisphosphate, inositol-1,4,5-trisphosphate, and diacylglycerol are involved in protein kinase C and calcium signaling. (**B**) Sphingomyelin and sphingosine-1-phosphate, sphingolipids synthesized from ceramide, are important membrane lipids and promote cellular proliferation, respectively. (**C**) Arachidonic acid products prostaglandin D2 and E2 can exhibit anti- and pro-inflammatory activities.

**Figure 3 metabolites-10-00339-f003:**
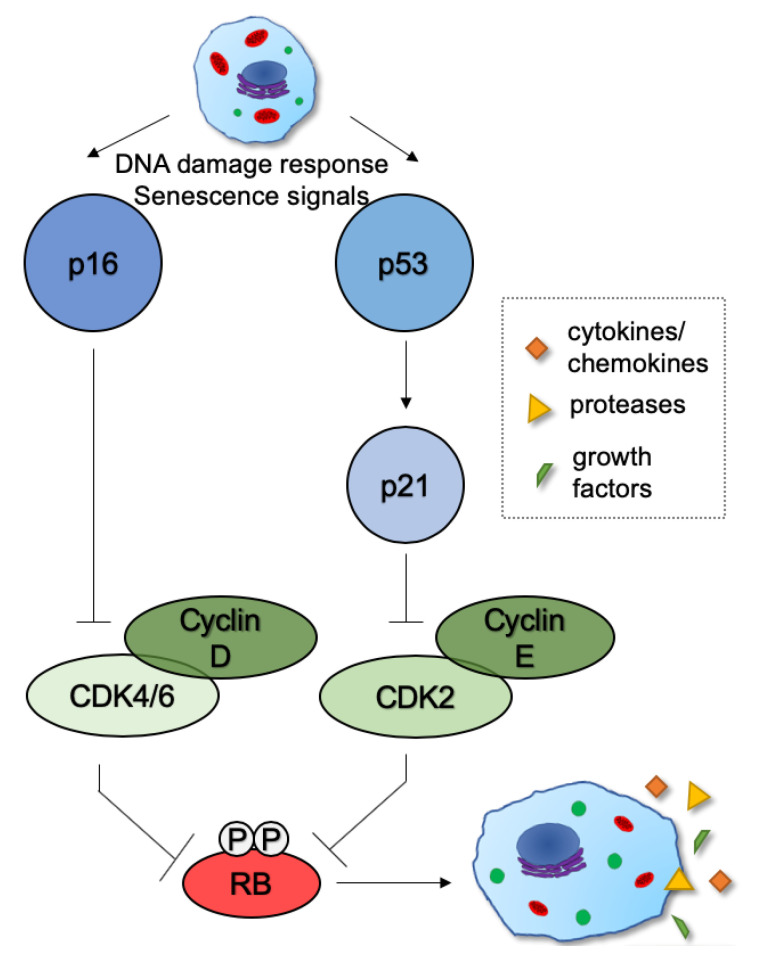
Mechanisms of establishing growth arrest. Growth arrest is established and maintained by two major tumor suppressor pathways, p53 and p16, during senescence. In proliferating cells, cyclin-dependent kinase (CDK)–cyclin complexes phosphorylate retinoblastoma (RB) proteins and promote cell cycle progression. In senescent cells, activated p16 and p21 inhibit these CDK–cyclin complexes from phosphorylating RB. Hypophosphorylated RB represses cellular proliferation.

**Figure 4 metabolites-10-00339-f004:**
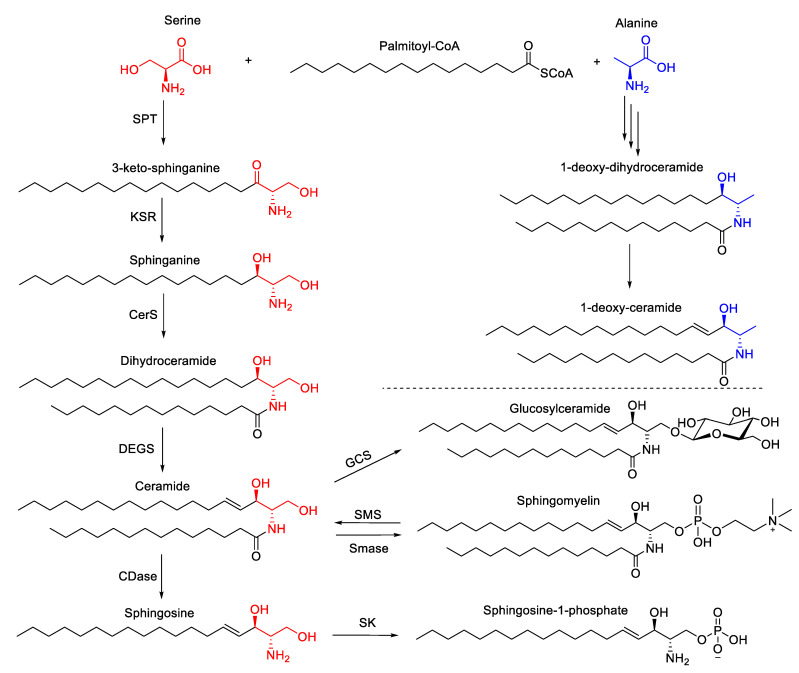
Simplified sphingolipid biosynthesis scheme. Condensation of serine (red) and palmitoyl-CoA produces 3-ketosphinganine, which is reduced to sphinganine and acylated, resulting in dihydroceramide. The trans double bond is introduced to the fourth carbon to yield ceramide, which serves as the precursor for more complex sphingolipids and sphingosine. Ceramide can be incorporated into glucosylceramide or sphingomyelin through the respective pathways. Ceramide can also be hydrolyzed to form sphingosine, which can then be phosphorylated to produce sphingosine-1-phosphate. Alternatively, alanine can be utilized (blue) rather than serine, resulting in the production of 1-deoxy-dihydroceramide and 1-deoxy-ceramide. Abbreviations: SPT (serine palmitoyl transferase); KSR (3-keto-sphinganine reductase); CerS (ceramide synthase); DEGS (dihydroceramide desaturase); CDase (ceramidase); GCS (glucosylceramide synthase); SMS (sphingomyelin synthase); (SMase (sphingomyelinase); SK (sphingosine kinase).

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
