# Peer review of "Lipid Players of Cellular Senescence"

_metabolites, 2020, doi:10.3390/metabo10090339_

Round 1

Reviewer 1 Report

The review presented by Millner and Atilla-Gokcumen entitle "lipid players of cellular senescence" is well written and the information provided will be helpful for readers interested in lipid metabolism and more specifically how lipid metabolism changes with aging and cellular senescence.

The manuscript includes and extensive revision of the literature but there are two main points that I recommend to be addressed in this review:

1. Authors have well structure the manuscript and discuss some lipid classes in the context of cellular sense, disease, health and aging. The main lipid classes discussed in this review are fatty acids, glycerolipids including DAGs and TAGs and sphingolipids. However, other lipid classes such as phospholipids with a specific focus on lysophospholipids have been shown to play important metabolic roles.

So I would suggest to the authors to include a discussion about phospholipids in the review. And also other lipid classes such as sterol lipids (steroids, Cholesterol and cholesterol esters).

2. Sex differences in plasma lipid composition and their contribution to sex differences in obesity, diabetes (T2D), atherosclerotic cardiovascular disease and neurodegenerative diseases were investigated in several large-scale population studies. Among the differences in lipid metabolism due to sex differences one of the most prominent is the differences in sphingolipids as recently reported by Huynh et al., (Cell Chemical Biology 26, 71-84.e74, doi:https://doi.org/10.1016/j.chembiol.2018.10.008 (2019)). Considering such differences, I encourage the authors to discuss the sex differences observed in lipid metabolism. And how these differences in the lipid metabolism between males and females is also affected by age due to changes in the hormonal status.

Author Response

Reviewer 1

The review presented by Millner and Atilla-Gokcumen entitle "lipid players of cellular senescence" is well written and the information provided will be helpful for readers interested in lipid metabolism and more specifically how lipid metabolism changes with aging and cellular senescence.

Response: We thank the reviewers for their time and interest in our review paper. We have incorporated their suggestions in our revised version. Please find our edits marked in blue in the manuscript file.

The manuscript includes and extensive revision of the literature but there are two main points that I recommend to be addressed in this review:

  1. Authors have well structure the manuscript and discuss some lipid classes in the context of cellular sense, disease, health and aging. The main lipid classes discussed in this review are fatty acids, glycerolipids including DAGs and TAGs and sphingolipids. However, other lipid classes such as phospholipids with a specific focus on lysophospholipids have been shown to play important metabolic roles. So I would suggest to the authors to include a discussion about phospholipids in the review. And also other lipid classes such as sterol lipids (steroids, Cholesterol and cholesterol esters).

Response 1: We thank the reviewer for pointing this out. We have included a section on phospholipids (lines 322-350).

Studies have shown correlations between sterol levels and different age-related diseases but we believe these studies do not fit the scope of our Review paper as we aimed to keep the focus on mechanistic studies on lipid involvement during replicative and stress-induced senescence. Hence, we included references for sterol-related metabolic changes during aging (lines 197-199) direct the interested readers to recent studies on this topic.

  1. Sex differences in plasma lipid composition and their contribution to sex differences in obesity, diabetes (T2D), atherosclerotic cardiovascular disease and neurodegenerative diseases were investigated in several large-scale population studies. Among the differences in lipid metabolism due to sex differences one of the most prominent is the differences in sphingolipids as recently reported by Huynh et al., (Cell Chemical Biology 26, 71-84.e74, doi:https://doi.org/10.1016/j.chembiol.2018.10.008 (2019)). Considering such differences, I encourage the authors to discuss the sex differences observed in lipid metabolism. And how these differences in the lipid metabolism between males and females is also affected by age due to changes in the hormonal status.

Response 2: We agree with the reviewer that plasma lipid composition can be greatly affected by sex and age and contribute to different diseases. In our review, we aimed to keep the focus on more mechanistic studies on lipid involvement in senescence. Thus, we believe a detailed discussion on the effects of sex and age on lipid composition is not within the immediate scope of this review. We have added a brief section (lines 197-199) to highlight these differences and direct the interested readers to recent studies on this topic.

Reviewer 2 Report

It is a well written review of a very interesting topic. There are a few points for the authors to consider/modify:

1) Recent research revealed that senescent cells displayed increased lipid aldehydes and upregulation of aldehyde detoxifying enzymes (Cell Death Discovery Vol. 3, 17075 (2017). The authors should include this aspect in the review. 

2) Is there evidence for gender difference in terms of how lipid affects cellular senescence? It would be interesting to the audience if the authors can discuss it.

Some minor concerns:

1) Figure 3 needs title.

2) Line 132, "senescence cells become enlarged...", does the authors mean " cells undergoing senescence..." or "senescent cells".

3) Line 166, please define GLB1.

4) Line 187, please define AA.

5) Line 329-330,"Sphingosine and S1P have shown to be involved in various cell fates",  does the authors mean "Sphingosine and S1P have shown to be involved in various processes regulating cell fates".

Author Response

Reviewer 2

It is a well written review of a very interesting topic.

Response: We thank the reviewers for their time and interest in our review paper. We have incorporated their suggestions in our revised version. Please find our edits marked in blue in the manuscript file.

There are a few points for the authors to consider/modify:

1) Recent research revealed that senescent cells displayed increased lipid aldehydes and upregulation of aldehyde detoxifying enzymes (Cell Death Discovery Vol. 3, 17075 (2017). The authors should include this aspect in the review.

Response 1: We thank the reviewer for pointing this out. We have included discussion on lipid aldehydes (lines 343-350).

2) Is there evidence for gender difference in terms of how lipid affects cellular senescence? It would be interesting to the audience if the authors can discuss it.

Response 2: We agree with the reviewer that plasma lipid composition can be greatly affected by sex and age and contribute to different diseases. In our review, we aimed to keep the focus on more mechanistic studies on lipid involvement in senescence. Thus, we believe a detailed discussion on the effects of sex and age on lipid composition is not within the immediate scope of this review. We have added a brief section in the (lines 197-199) to highlight these differences and direct the interested readers to recent studies on this topic.

Some minor concerns:

1) Figure 3 needs title.

A title has been added to Figure 3, line 157

2) Line 132, "senescence cells become enlarged...", does the authors mean " cells undergoing senescence..." or "senescent cells".

This has been corrected to “Senescent cells…”, now line 136.

3) Line 166, please define GLB1.

GLB1 has been replaced with full gene name: “galactosidase beta 1”, line 166.

4) Line 187, please define AA.

AA is the isoform of the growth factor. This has been clarified, line 188.

5) Line 329-330,"Sphingosine and S1P have shown to be involved in various cell fates",  does the authors mean "Sphingosine and S1P have shown to be involved in various processes regulating cell fates".

This has been clarified to “…various processes regulating cell fates,..” line 362.